# Impact of Lung Microbiota on COPD

**DOI:** 10.3390/biomedicines10061337

**Published:** 2022-06-06

**Authors:** Cristina Russo, Valeria Colaianni, Giuseppe Ielo, Maria Stella Valle, Lucia Spicuzza, Lucia Malaguarnera

**Affiliations:** 1Section of Pathology, Department of Biomedical and Biotechnological Sciences, University of Catania, 95123 Catania, Italy; cristina.russo@unict.it (C.R.); colaianni.valeria@gmail.com (V.C.); 2Department of Clinical and Experimental Medicine, University of Catania, 95123 Catania, Italy; giuseppeielo@hotmail.it (G.I.); lucia.spicuzza@unict.it (L.S.); 3Laboratory of Neuro-Biomechanics, Section of Physiology, Department of Biomedical and Biotechnological Sciences, School of Medicine, University of Catania, 95123 Catania, Italy

**Keywords:** COPD, microbiota, inflammation, lung, dysbiosis

## Abstract

There is a fine balance in maintaining healthy microbiota composition, and its alterations due to genetic, lifestyle, and environmental factors can lead to the onset of respiratory dysfunctions such as chronic obstructive pulmonary disease (COPD). The relationship between lung microbiota and COPD is currently under study. Little is known about the role of the microbiota in patients with stable or exacerbated COPD. Inflammation in COPD disorders appears to be characterised by dysbiosis, reduced lung activity, and an imbalance between the innate and adaptive immune systems. Lung microbiota intervention could ameliorate these disorders. The microbiota’s anti-inflammatory action could be decisive in the onset of pathologies. In this review, we highlight the feedback loop between microbiota dysfunction, immune response, inflammation, and lung damage in relation to COPD status in order to encourage the development of innovative therapeutic goals for the prevention and management of this disease.

## 1. Introduction

Chronic obstructive pulmonary disease (COPD) is characterised by persistent inflammation of the lower airways, dysfunction of mucociliary activity, and emphysematous destruction of the lungs culminating in irreversible airflow limitation [1]. The incidence of COPD has increased worldwide, and it is the principal cause of morbidity and mortality [2]. The common symptoms of the disease include dyspnoea, cough, sputum, and altered respiratory composition resulting from chronic exposure to cigarette smoke (CS) or pollutants [3]. In fact, CS affects the microbial composition mainly through lack of oxygen and increased formation of bacterial biofilms [3,4]. Inhalation of CS results in the activation of epithelial and immunomodulatory cells, including neutrophils, macrophages, and T cells [5]. The respiratory tract is one of the main entrances for microorganisms into the human body; thus, it plays a key role in the composition of the microbiota. The airway microbiota is finely organised, building a protective environment for the host and a hostile condition for pathogenic agent colonisation, ensuring a balanced relationship between microbiota and human host signals [3]. Inhaled microorganisms find the ideal habitat to develop in the lung, and they are also able to reduce factors that regulate their elimination. Respiratory diseases alter the right intake/removal of germ balance [6]. This different colonisation, disrupting the immunomodulatory effect of the microbiota, may induce lung dysfunction, contributing to pathogenesis and the clinical course of the disease. Therefore, alterations in the local microbial composition of the respiratory system are associated with inflammation of the airways and the progression of COPD [3]. Lung microbiota composition changes during the progression of the disease. In addition, numerous factors such as environment, age, lifestyle, diet, and pharmacological treatment further contribute to dysbiosis [6], an imbalance in the microbiota composition caused by variations in local conditions. Therefore, lung dysbiosis leading to exacerbations of COPD could have a strong impact on health status, exercise capacity, lung function, and mortality [7]. Increasing evidence shows that the dysbiosis of the lung microbiota could play an important role in COPD development and progression [8,9]. Moreover, it has not yet been clarified whether the dominant pathogenic bacteria can inhibit the growth of resident bacteria in the lungs and promote disease. This review aims to provide some mechanistic insights to determine whether there is a feedback loop between microbiota dysfunction, inflammation, and lung damage in order to encourage the development of innovative therapeutic goals for the prevention and treatment of COPD and to avoid its exacerbation.

## 2. Lung Microbiota

Traditionally, the lung was considered a sterile organ; therefore, bacterial colonisation was taken into account in the context of lung disease [10]. The lung microbiota has been categorised through metagenomic next-generation sequencing and rRNA gene sequencing of DNA extracted from samples collected via sputum, lung tissues, or bronchoalveolar lavage fluid (BALF) [11]. These studies revealed that the lungs of healthy subjects are not microbe-free. In fact, the lung has its own microbiota consisting of bacteria, viruses, and fungi [12]. However, the number of microorganisms inhabiting the lungs is significantly lower than those inhabiting the gut. Microbiota composition depends on a multitude of factors such as bacterial migration (microaspiration, inhalation of microorganisms, etc.), microbial elimination (cough, innate and adaptive immunity, etc.), and local conditions (nutritional availability, temperature, O_2_ tension, local microbial competition, the activity of inflammatory cells, etc.) [7]. The upper respiratory tract has a microbiota more similar to the oral cavity, and it can be strongly influenced by environmental factors [10], whereas in the lower respiratory tract the microbiota seems only slightly influenced by the microbiota from other mucosal surfaces. Bacteria present in the lower respiratory tract mainly include *Megasphaera*, *Streptococcus*, *Pseudomonas*, *Fusobacterium*, and *Sphingomonas* [13]. During pulmonary illnesses, the several airway sections can influence each other, in a dysbiosis state [14]. Therefore, several components of healthy lungs do not have a similar habitat. Another important component of the lung microbiota is viruses. The lungs can be exposed to viruses during respiratory activity [15]. Even fungi may play a role in immune modulation and preserve mucosal tissue impairment at the physiological state. As regards fungi, in the lungs, *Aspergillus, Cladosporium, Penicillium*, and *Eurotium* genera dominate [16,17]. A recent study has reported that the fungal microbiota in stable COPD patients was different in eosinophilic and non-eosinophilic inflammatory phenotypes. *Aspergillus* and *Bjerkandera* were significantly more abundant in COPD patients with eosinophilic inflammation, while non-eosinophilic COPD subjects had higher relative amounts of *Rhodotorula* and *Papiliotrema* [18]. This study strongly suggested that also fungal microbiota was linked with a particular airway inflammatory environment, and it demonstrated the connections of differential microbiota features in particular phenotypes between subjects with asthma and COPD.

Lung microbiota status is preserved and controlled by the host’s immune system. Therefore, changes in the microbiota modulate host immunity and metabolism, which, in turn, alters growth conditions for bacteria in the airways, promoting further microbiota perturbation and perpetuating a cycle of inflammation and dysbiosis. The lung microbiota, besides being inherently variable, is also personal, with large individual variability. Proper development of the lung microbiota is even before birth and in the first years of life, changing with age, diet, living environment, CS, and the use of antibiotics [19]. Exposure to antibiotics early in life may also be a causal factor in the development of chronic inflammation, as antibiotic treatment can alter and reduce microbial diversity [20]. In addition, the integrity of the composition and correct maturation of the microbiota in the first period of life can prevent certain lung diseases or may, in the case of lung microbial imbalance, cause various pathological states [8].

## 3. Impact of Lung Microbiota on Immunity and Inflammation

The immune system consisting of innate and adaptive components regulates host homeostasis by supporting and restoring the function of tissues altered by external disturbances such as environmental and microbial influences [21]. At the same time, the microbiota, promoting and calibrating innate and adaptive immunity, provides resistance to invasion from respiratory pathogens. The lung microbiota is constantly in contact with the air, competes with invading microbes, regulates and influences the environment, enhancing the action of the immune system against pathogens [6,22].

The immune system and the microbiota cooperate, creating a strong synergism to promote protective responses and support tolerance, a phenomenon referred to as resistance to colonisation [23]. After a change in the composition and function of the microbiota, pathologies can occur. In the airway epithelium, pattern recognition receptors (PRRs), dendritic cells (DCs), and alveolar macrophages are located, elements able to recognise and differentiate microbial molecules as dangerous or not dangerous. Damage associated with the respiratory microbiota or pathogens leads to the induction of tolerance through Toll-like receptors (TLRs) that belong to single-pass transmembrane receptors expressed in innate immune cells. The respiratory microbiota, interacting with the PPRs of the airways and phagocytic cells, develops immunological tolerance, preventing exaggerated inflammatory responses [24]. For example, against *Pseudomonas aeruginosa,* the microbiota stimulates the production of Immunoglobulin A (IgA) through TLRs, to improve host defence, while against *Escherichia coli*, TLR4 stimulates alveolar macrophages to release inflammatory cytokines (Figure 1) [25].

TLRs activate various phases in the inflammatory reactions that help to eliminate the invading pathogens and coordinate systemic defences [26]. It has been found that treatment with flagellin, the main component of bacterial flagella and the TLR5 ligand, can control bacterial infection in CS-exposed mice and limit lung inflammation and remodelling. Moreover, systemic administration of flagellin induces rapid production of T-helper cells (Th)17 cytokines through the activation of DC and type 3 innate lymphoid cells [27]. The increased susceptibility to infection during COPD is linked to a defect in interleukin (IL)-22 production related to an altered innate immune response [28]. Both IL-17 and IL-22 promote the recruitment of neutrophils, the synthesis of antimicrobial peptides, and the expression of tight junction molecules (Figure 1) [28]. It has been reported that, in response to bacteria, both in COPD patients and mice chronically exposed to CS, there is a defective production of IL-22, whereas IL-17 production is only altered after infection by *Streptococcus pneumoniae* [7].

Nevertheless, TLRs inducing cytokines such as IL5, IL6, IL13, and interferon-gamma (IFN-γ), recruit immune cells into the lungs and exacerbate inflammation [29]. Chronic lung inflammation induced by IL-13 promotes emphysema via the stimulation of alveolar macrophages to release matrix metalloproteinase (MMP)-12 and induces airway space enlargement [30]. Another study using IL-13 overexpression in transgenic (TG) mice described the induction of MMP-2, MMP-9, MMP-12, MMP-13, and MMP-14; cathepsins such as B, S, L, H, and K, involved in emphysema development and increased lung inflammation [31].

The lung microbiota presents a relative deficiency in the phyla *Firmicutes* and *Proteobacteria* in IL-13 TG mice [32]. Differential exposure to the lung microbiota is associated with differential cellular immune responses. The most commonly involved bacteria appear to be *Pseudomonas* and *Staphylococcus*, both components of healthy lung microbiota. Exposure to *Pseudomonas* and *Lactobacillus* in mouse models of chronic lung inflammation enhanced Th responses, while *Proteobacteria* were associated with severe TLR2-independent airway inflammation and lung immunopathology [33]. The nucleotide-binding oligomerisation domain protein (NOD) is another PRR required for interaction with the microbiota. In particular, during infection, it has been reported that NOD2 promotes the macrophage response following binding with the peptidoglycans of the microbiota [34]. In COPD, another factor involved in this interaction is the mucus. On the one hand, the microbiota influences the production of mucus and the antimicrobial and immunomodulatory peptides secreted by the epithelium of the airways, such as cathelicidin and β-defensin, two families of cationic proteins involved in shaping the composition of the microbiota [24,35]. On the other hand, the mucus accumulating on the lining of the lung promotes microbiota alterations, as well as macrophage and neutrophil proliferation, inducing immune-mediated airway impairment and severely limiting lung function [36]. To prevent chronic inflammation in a healthy state, the lung microbiota plays a fundamental role in shaping pulmonary immune tolerance. The lung microbiota can prevent the growth and spread of invasive microbes and regulate immune tolerance by recruiting and activating Treg cells, macrophages, and DCs (Figure 1) [22]. Macrophages are an important defence component of airways, which serve as sentinels working to contrast external antigens. Macrophages endowed with different polarisation could be distinguished into M1-like, identified as pro-inflammatory nature, and the anti-inflammatory M2-like with phagocytic capacity and ability to release IL10 [37,38]. Macrophage polarisation may be linked with COPD [39]. A recent study showed that, in alveolar macrophages, CS promotes M2 polarisation and the increase in IL4 inducing MMP12 release, which is involved in emphysema. M2 macrophages also induce the release of IL8 involved in neutrophil elastase and Mucin 5AC (MUC5AC) expression, which lead to hypersecretion of mucus and possible obstruction of the airway (Table 1) [40]. Bacterial metabolites, such as butyrate, induce M2 polarisation and suppress M1 polarisation of macrophages both in vivo and in vitro. Butyrate inhibits the expression of nuclear factor kappa-light-chain enhancer of activated B cells (NF-κB), p38, and extracellular signal-regulated kinase 1 (ERK1) in respiratory syncytial virus-infected murine macrophages, demonstrating the involvement of NF-κB, p38, mitogen-activated protein kinase, and ERK1 signalling pathway in its activity. It is well-known that NF-κB and ERK regulate inflammation, decreasing the release of IFN-γ and IL-17, as well as increasing IL-5 levels. NF-κB and ERK are involved in the modulation of nitric oxide synthase (iNOS) and the expression of various cytokines in macrophages [41]. Therefore, it is possible that NF-κB, p38, and ERK1 are target molecules of butyrate in the regulation of pro- and anti-inflammatory cytokine production in macrophages. It has recently been found that supplementation of *Clostridium butyricum* prevents inflammation aggravation and dysregulates the immune response characterised by greater M2 polarisation of pulmonary macrophages and decreased release of IFN-γ and IL-17 as well as increased IL-5 levels [42]. Additionally, an increase in *Pseudomonas* and *Lactobacillus* genera in a murine COPD model has been described, while there was a reduction in *Prevotella*. Mice depleted or devoid of microbiota exhibited an improvement in lung function, reduced inflammation, and lymphoid neogenesis [33]. This correlated with studies in humans that showed that *Prevotella* is a common bacterium in the airways of healthy subjects, compared with COPD patients [43]. Similarly, *Pseudomonas* and *Lactobacillus* increased in patients with COPD, and disease severity (Table 1) [44]. These data indicate that host–microbial cross-talk promotes immune system activity by triggering the inflammasome and causing an increase in inflammatory cytokines, leading to the chronicity of inflammatory lung disorders. Therefore, the microbiota influences and is influenced by both immunity and disease. During chronic inflammation, the lung habitat becomes unstable, and its species composition switches from healthy to a pathogenic condition [45]. Genetic problems and dysbiosis often contribute to the development of chronic lung disease. Growing evidence suggests that the respiratory microenvironment can be altered by metabolites or toxins released by the microbiota. Of course, it is not the effect of a single metabolite or bacterial species, but the interconnected events that lead to the development of diseases [46].

## 4. Impact of Lung Microbiota on Oxidative Stress in COPD

During inflammation in COPD, alveolar macrophages and neutrophils are able to generate reactive oxygen species (ROS) as well as nitrogen species [52]. The increased levels of both species induce an oxygen depletion to the surrounding mucus, transforming the lung microenvironment from an aerobic environment to an anaerobic environment, leading, in turn, to an increase in anaerobic bacteria. *P. aeruginosa* is among the anaerobic bacteria present in the lung, which belongs to *Gammaproteobacteria*. They are able to live in both aerobic and anaerobic conditions, becoming pathogenic in certain circumstances. In fact, when bacterial diversity decreases in the lungs, in most cases, *P. aeruginosa* increases [53]. Once *P. aeruginosa* infection is established, it correlates with high mortality rates, because it leads to acute lung injury as well as respiratory distress syndrome, with treatment complications [54] (Figure 2).

When oxygen levels are very low, *P. aeruginosa* adopts aerobic respiration, which is important for its colonisation and pathogenesis. In anaerobic conditions, *P. aeruginosa* performs its metabolic activity through the denitrification pathway, which consists of the use of nitrogen oxides as final electron acceptors. Denitrification comprises four steps leading to the reduction of NO3¯ to N2; each molecule of these steps is catalysed by a reductase [55] (Figure 2). However, in order for *P. aeruginosa* to perform denitrification, an abundance of nitrogen has to be available in the lung [56].

Defective phagocytosis was reported in COPD macrophages associated with an alteration in the mitochondrial function unable to regulate mitochondrial ROS (mROS) production [57,58]. It is known that ROS can cause damage to various molecules, such as proteins, DNA, as well as mitochondrial DNA, but mROS has a positive aspect, as they represent an advantageous mechanism by which, following phagocytosis, macrophages eliminate bacteria [59]. In healthy individuals, the mROS concentration is controlled by scavenged enzymes to prevent cell damage [60]. In the macrophages of COPD patients, mROS increase incessantly because these cells are no longer able to regulate this production. Excessive levels of mROS prevail over the antioxidant enzymes, causing dysregulation (Figure 1) [61]. Investigating the mitochondrial functional alterations could represent a potential therapeutic approach.

## 5. Lung Microbial Dysbiosis and COPD

The lung is a vulnerable organ, as it communicates directly with the external environment, and exposure to microorganisms, allergens, and pollutants can alter the composition of the microbiota. Bacterial persistence can be influenced by several agents such as changes in oxygen concentration, pH, available nutrients, and temperature. Alteration in these factors can cause microbial dysbiosis, which can lead to the development of several diseases such as COPD [61]. Analysis of COPD patients and healthy controls showed differences in the composition of the microbiota (Table 1) [62]. More specifically, in a very recent study, it was found that in the healthy controls at the phylum levels *Firmicutes* (34.01%), *Bacteroidetes* (26.01%), and *Proteobacteria* (23.09%) were prevalent [63]. The dominant bacterial phylum in acute exacerbations of COPD (AECOPD) group was *Proteobacteria* (30.29%), followed by *Firmicutes* (29.85%) and *Bacteroidetes* (14.02%). In the stable COPD group, the major phylum was *Firmicutes* (31.63%), followed by *Bacteroidetes* (28.94%) and *Proteobacteria* (19.68%). In the recovery group, the major phyla were *Firmicutes* (44.04%), *Proteobacteria* (21.94%), and *Bacteroidetes* (13.35%) [63], while at the genera level in the lungs of COPD patients, the most prevalent detected were *Streptococcus*, *Staphylococcus*, *Pseudomonas*, *Prevotella*, *Veillonella*, and *Gemella* [64]. In another study, the most abundant phyla identified were *Firmicutes*, *Proteobacteria*, *Bacteroidetes*, and *Actinobacteria*, containing 97% of all sequences both in a stable state and during exacerbations. Nevertheless, *Proteobacteria* was relatively more dominating in samples collected during exacerbations, compared with those collected in stable conditions. By contrast, *Streptococcus*, *Rothia*, *Prevotella*, *Veillonella*, and *Haemophilus* together contained 68% of all sequences under stable conditions, and during exacerbations, they were the most abundant genera [47]. Therefore, lung diseases may cause microbiota imbalance, the ineffectiveness of germ elimination, and airway alterations. These modifications promote the formation of microbial niches that encourage the growth and the increase in common anaerobic commensals such as *Prevotella* and *Veillonella* [65]. These commensals induce inflammation in the airways, supported by lymphocytes and neutrophils [33]. Studies have also reported a higher abundance of *Streptococcus* in COPD patients, compared with healthy controls [66]. In COPD, the microbiota diversity in the samples depends on the type of treatment received, disease severity, and inflammation. A study aimed at comparing the microbial diversity from nasal and oral cavities of healthy patients, including the risk factors of tobacco smoke-associated COPD (TSCOPD) and biomass smoke-associated COPD (BMSCOPD), showed significant differences in the microbiota in healthy vs. COPD and/or, TSCOPD vs. BMSCOPD subjects [67]. In particular, *Actinomyces*, *Actinobacillus*, *Megasphaera*, *Selenomonas*, and *Corynebacterium* were significantly higher in COPD subjects. *Acinetobacter* was higher in the oral samples of TSCOPD subjects, while *Gallibacterium* and *Methanosaeta* were higher in the nasal samples of BMSCOPD subjects. The presence of a specific microbial community of bacteria in BMSCOPD suggested a difference in the pathophysiology of BMSCOPD, compared with TSCOPD, and also explained the clinical phenotypic differences between these groups (Table 1) [67]. When comparing the microbiota diversity of COPD and other chronic lung diseases characterised by a rapid decline in lung function, such as idiopathic pulmonary fibrosis (IPF), it has been reported that the most abundant genera were *Streptococcus* sp., *Rothia*, *Veillonella*, and *Prevotella* [68]. The presence of a specific *Streptococcus* sp. or *Staphylococcus* sp. was strongly associated with acute exacerbation, disease progression, and survival [48].

## 6. Lung Microbial Dysbiosis and COPD Exacerbation

COPD disease development is characterised by exacerbation that is associated with amplified inflammation and reduced lung function [67].

Exacerbation consists of cough aggravation, dyspnoea, fever, and sputum colour or viscosity change [69,70]. Exacerbations are usually caused by viral and bacterial respiratory infections with common pathogens such as *Haemophilus influenzae*, *Moraxella catarrhalis*, *P. aeruginosa*, and *S. pneumoniae* [71]. These pathogens lead to exacerbations of the disease, which have a strong impact on health status, exercise capacity, lung function, and mortality [72,73]. Non-typeable *Haemophilus*, a Gram-negative coccobacillus, which lacks a polysaccharide capsule, is an important cause of COPD exacerbations and comorbidity [71]. *H. influenzae* membrane protein can induce mucin production in COPD mice and cultured human epithelial cells [74]. Sputum analysis from different exacerbation periods had the same strain of *H. influenza*, suggesting that the pathogens stay within the host and increase in abundance during or before exacerbations. An important pathological feature of COPD is emphysema. CS is the most predominant risk factor for COPD development. Chronic exposure to CS has been shown to alter the microbiota in the upper respiratory tract [43]. However, Segal et al. did not find any difference in the microbial community between smokers, non-smokers, and early COPD; however, several studies reported differences in the airway microbiota between healthy and tobacco smoke-associated COPD individuals [20,49]. Microbial alterations in ex-smoker COPD patients were associated with strong airway inflammatory conditions, as shown by the enhanced pro-inflammatory marker expression from ex-smoker COPD patients’ sputum. *Haemophilus* was relatively abundant in COPD ex-smokers versus smokers [4,5]. It was also shown that *Megasphaera*, a member of the lung microbiota correlated with a reduced expression of host inflammatory condition, could potentially modify the airway inflammation caused by *Moraxella* and *Haemophilus*. The beneficial effects of *Megasphaera* on inflammation are due to the production of short-chain fatty acids that inhibit cytokine production [75]. Smoking and *Moraxella* infection are both associated with emphysema. Increased endothelial monocyte-activating protein-II (EMAPII), MMP-9, and MMP-12 levels are the primary cause of the development of emphysema by *Moraxella* [76]. It is not yet clear if smoking contributes to lung disease or is a side effect of the induced *Moraxella* genetic mutations (Table 1) [62]. In COPD subjects, *Haemophilus* represent 25% of the airway microbiota, but they were at a significantly lower proportion in healthy conditions. *Moraxella* contributed 3% to the lung microbiota in COPD patients. The genera *Haemophilus* and *Moraxella* were strongly associated with gene expression, host immunity, and inflammation. *Haemophilus* was positively correlated with neutrophilia, while interferon signalling was more strongly linked to *Moraxella*. In addition, *Haemophilus* was significantly associated with host factors both in stable conditions and during exacerbations. This supported an earlier study in which *Haemophilus* was considered a constant airway coloniser, while *Moraxella* was an opportunist in COPD exacerbations (Table 1) [77]. There are currently oral vaccines for *H. influenzae* infection. Although it has been demonstrated that oral vaccination can prevent acute exacerbation of COPD, it cannot significantly decrease the incidence and severity of acute exacerbation of COPD development [78]. Several studies reported differences between stable and exacerbated conditions in COPD, raising the question of whether bacterial colonisation in the lungs is responsible for future exacerbations [47]. However, the results are somewhat contradictory. A study reported that lung dysbiosis caused exacerbations and that differences in baseline microbiota could help to explain the frequency of the exacerbation type [70]. In contrast, in other investigations, a significant airway host–bacterial interaction associated with COPD inflammation and exacerbations was found. A study on stable COPD patient BALF samples assumed that *Pseudomonas* played a role during exacerbations, while *Streptococcus* and *Rothia* protected against exacerbations [50]. Another study also showed that *Pseudomonas*, *Selenomonas*, and *Anaerococcus* increased in patients with frequent exacerbations, compared with patients without exacerbations [51]. In COPD patients with and without exacerbation, the most prevalent genera were *Streptococcus*, *Veillonella*, *Prevotella*, and *Gemella.* The relative amounts of different taxa exhibited a large variation between individuals and samples. The individual variations in the lower airway microbiota in COPD patients overcame the group differences between infrequent and frequent exacerbations. The abundance of *Streptococcus* appeared higher in the participants without later exacerbation than in COPD patients with exacerbation. This supports the hypothesis of the protective capacities of *Streptococcus* [51]. Contrary to these findings, in a cohort study, no significant association was found between the lung microbiota in stable COPD patients and COPD exacerbation frequency. Furthermore, no association between *Pseudomonas* was found, and more frequent exacerbations were observed [79]. These findings challenge the few existing published reports. This was probably due to the fact that the researchers used a larger sample size, prospective follow-up of exacerbations, and protected sampling of the lower airways such as bronchoscopy, which is superior to sputum sample collection. Sputum samples are easily contaminated by microbes from the upper airways, making the interpretation of the lower airway microbiota difficult. Some studies have recently revealed that the sputum microbiome community changed in stable COPD patients with different risks of exacerbation [80]. The bacterial diversity was significantly reduced in high-risk exacerbator (HRE) subjects, compared with low-risk exacerbators (LREs). The dominant phyla in the sputum microbiome were *Firmicutes*, *Actinobacteria*, and *Proteobacteria*, and their abundance was similar in both the HRE and LRE groups. The proportions of *Gemella morbillorum*, *Streptococcus gordonii*, and *Prevotella histicola* were increased in the LRE group, compared with the HRE group, while the opposite was found for the *Neisseria* species. There was no microbial diversity or different proportions of bacteria in groups with different lung function levels. Bacterial cooperative regulation was different in the diverse risks of COPD exacerbation subgroups. In particular, in the HRE group, there was an altered bacterial cooperative network. The authors suggested that lung microbiota alteration can be an important factor in exacerbation risk in stable COPD patients.

In a multicentre COPD cohort study, Bouquet et al. evaluated both viral and bacterial microbiotas in sputum samples of severe COPD patients, collected following medical examinations during acute exacerbation and follow-up (Table 1) [81]. They found a comparable viral and bacterial microbiota in patients from different countries; however, their microbiota variability was substantially different and did not influence exacerbation frequency. According to the level of peripheral blood and/or sputum eosinophil count, COPD patients can be distinguished into non-eosinophilic and eosinophilic inflammatory phenotypes. Patients with low blood eosinophil count display *Proteobacteria* dysbiosis with a dominance of *Haemophilus*, *Moraxella*, and *Pseudomonas genera*. These patients show lung function, chronic bronchitis symptoms, exacerbations, and lastly, increased mortality. In addition, *Proteobacteria* dominance (predominantly *Haemophilus*) is connected with increased neutrophilic inflammatory responses and neutrophil extracellular trap (NET) formation. As a consequence, the production of neutrophilic inflammatory markers, such as neutrophil elastase, myeloperoxidase, and matrix metalloproteinases, promotes disease progression in emphysema and reduces survival. In contrast, patients with eosinophilic inflammatory phenotypes exhibited a positive relationship with *Firmicutes* dominance (predominantly *Streptococcus*) and had a milder disease and frequent exacerbations [82]. This study suggested the identification of microbiome-based subtypes of COPD associated with clinical phenotypes.

Virus infections such as human rhinovirus, influenza virus, and coronavirus were associated with acute exacerbation events but had a lower significance in regard to the association with exacerbation frequency. *Moraxella* and *Haemophilus* were linked to exacerbation events, while *Pseudomonas* and *Staphylococcus* were associated with exacerbation frequency. This supports a key role of the microbiota in sensitising the COPD lung to acute exacerbations; however, viral infections alone do not sensitise the lung to exacerbations as much as the bacterial microbiota.

Furthermore, COPD patients can be susceptible to fungal infections [83]. Fungi are present in the lung in a lower number, compared with bacteria, and their specific role in COPD remains unclear [62]. Studies reported associations between fungal perturbations and worse clinical outcomes. For instance, *Aspergillus* and *Candida albicans* infections in COPD are associated with increased symptoms and acute exacerbations [84]. Oral steroids and long-term inhaled corticosteroids, used during exacerbations, alter the host’s immune system and may predispose patients to fungal-associated disease [84]. However, in terms of COPD, only a single study evaluated the role of the mycobiome in a specific patient subgroup with Human Immunodeficiency Virus (HIV), observing that the key perturbation to airway mycobiota was an increased number of *Pneumocystis jirovecii* [85]. This opportunistic fungal pathogen in COPD patients is also associated with severe airflow obstruction [86]. In contrast, a multicentre COPD study reported that the respiratory mycobiota is characterised by specific fungal genera associated with very frequent exacerbations and increased mortality. This study did not detect any *Pneumocystis fungi* in COPD cohorts. Two clinically important COPD groups were identified, one characterised by significant symptoms and *Saccharomyces* species, and the other group was characterised by frequent exacerbations and higher mortality, as well as *Aspergillus*, *Penicillium,* and *Curvularia*
*genera*. This last group showed a systemic immune response with an increase in specific IgE against these fungi. The authors affirmed that the COPD mycobiome could be used to identify high-risk patients, with worse two-year survival following an acute exacerbation [83].

## 7. Gut–Lung Axis and COPD

Emerging evidence has shown that gut microbiota also plays a role in lung disease, including COPD. Gut and lung microbiota originate from the same germ layer and share similar anatomical structures [87]. Thus, this bidirectional gut–lung cross-talk, termed the gut–lung axis, allows endotoxins and metabolites released by the gut microbiota to reach the lungs through the bloodstream, and for those of the lung microbiota to reach the gut.

The balance between the gut and lung microbiota is essential for maintaining the health of the host. Although it is not still completely understood if a gut microbiota dysbiosis is a cause or a consequence of lung disease, it is well-known that the gut microbiota plays a role in influencing the immune response of the gut and other distal organs, including the lungs [88]. Mucosal epithelial integrity impedes the entrance of bacteria and their products into the systemic circulation. Impaired intestinal integrity enhances the permeability of the mucosal barrier, facilitating bacterial translocation [89]. It has been observed that increased gastrointestinal permeability to bacterial entrance promotes systemic inflammation in AECOPD [63]. Therefore, dysfunctions in gut microbiota could critically affect lung diseases [90].

On the other hand, patients affected by COPD have a higher probability of also presenting chronic gastrointestinal tract diseases [91].

## 8. Probiotics and Their Therapeutic Potential in COPD Patients

The use of probiotics has been proposed as a possible treatment for COPD patients, given that the therapies available are still only palliative [1]. Probiotics are live microorganisms that, administered in adequate amounts, confer health benefits on the host. To date, the knowledge of the lung microbiota and the mechanisms of action by which pathogen inhibition is assessed have improved probiotic use for lung infections (Table 2). It is known that they regulate the respiratory immune response through several signalling pathways. It has been reported that *Bifidobacterium* modulates humoral and cellular immune responses [92], whereas *E. coli* strains such as Nissle 1917 (ECN) could decrease the recruitment of inflammatory cells in the lungs [93]. Moreover, *Lactobacillus plantarum*, which acts on Treg, is able to induce immunosuppression, decreasing the number of macrophages and neutrophils in the lungs and cytokine levels such as IL-6 and TNF-α [94]. Intranasal administration of probiotics stimulates immune responses, protecting the host from infection in the airways [95]. In an experimental mouse model, *Lactobacillus casei* stimulated the migration of immune cells, influencing cytokine expression and decreasing pathogens and pneumococci in the lung [96]. Emerging data also propose that daily intake of probiotics such as *L. casei* increases natural killer cell action in COPD smoker patients (Figure 3) [97].

Collectively, these data suggest that identifying probiotic organisms capable of reducing lung infections may contribute the prevention of disease development.

## 9. Concluding Remarks and Future Directions

COPD is a heterogeneous and multifactorial disease characterised by various phenotypes. Overall, this report suggests that the changes in microbiota composition contribute to the pathogenesis of COPD inducing and/or depending on the heterogeneous phenotypes of the disease with an important impact on exacerbation risk, quality of life, and even survival. It is still unclear whether changes in the microbiome are the cause or consequence of airway inflammation, restriction of airflow, and destruction of pulmonary and bronchial parenchyma. We described the relationship between the appearance of symptoms of exacerbation and the acquisition of new bacterial strains, but this change in bacterial flora only partially explains the appearance of exacerbations. During exacerbations, some genera enrich their abundance, while others do not change significantly. Therefore, exacerbations correlate with an increased presence of isolated genera, but also with parallel modifications in microbiome composition as a whole, which lead to an intensification of inflammatory response.

Remarkably, the gut–lung axis is strongly linked with COPD. The gut and lung microbiota influence COPD onset and progression. Furthermore, metabolites secreted by the gut microbiota are able to suppress inflammatory responses. Diet and nutritional status have a high impact on the gut microbiota, influencing its profile and interaction with the immune system [88]. Likewise, in COPD patients, malnutrition and nutrient deficiency such as low levels of vitamins and antioxidant is frequent and commonly leads to worse consequences [98]. These data highlight the strong anti-inflammatory power of the microbiota in COPD. Modification of both lung and intestinal microbiota, especially in the early stages of life, with the use of a specific diet and correct lifestyle, as well as vitamin and/or probiotic supplementation, can be a protective factor for the onset and progression of respiratory diseases. The lung microbiota is closely linked to the host’s immune status; thus, succeeding in identifying microbial populations could be useful in predicting the effectiveness of different treatments. The host’s interaction with the microbiota implies that a balance between stimulatory and regulatory signals would allow for the development of immunity without compromising the capacity of the host to maintain tolerance to innocuous antigens. The modalities of interaction between host and immune response must be better investigated in order to look for new therapeutic interventions able to positively modulate this response.

## Figures and Tables

**Figure 1 biomedicines-10-01337-f001:**
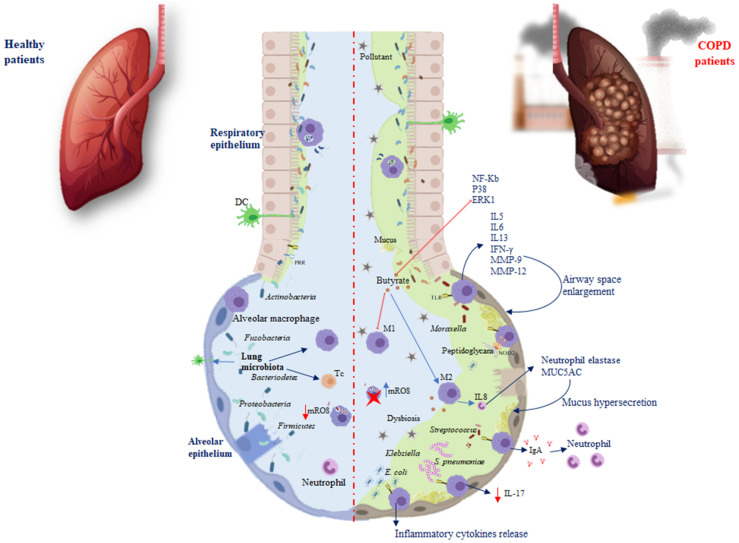
Differences in microbiota immunoregulation in the alveolar epithelium of healthy individuals and COPD. Red arrows show inhibition processes, while blue arrows show activation processes. Abbreviations: pattern recognition receptors (PRRs); Toll-like receptors (TLRs); Treg cells (Tc); macrophages (M); DCs (dendritic cells); Immunoglobulin A (IgA); Interleukin (IL); interferon-gamma (IFN-γ); matrix metalloproteinase (MMP); mitochondrial reactive oxygen species (mROS); nuclear factor kappa-light-chain enhancer (NF-Kb); P38 mitogen-activated protein kinases (P38); extracellular signal-regulated kinase (ERK1). Red arrow shows an inhibition process; red cross indicates an inactivated cell.

**Figure 2 biomedicines-10-01337-f002:**
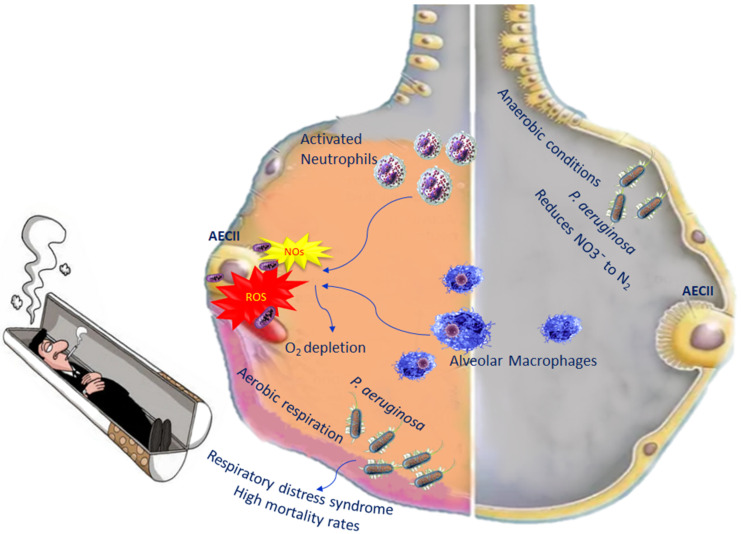
Impact of *P. aeruginosa* on oxidative stress in COPD.

**Figure 3 biomedicines-10-01337-f003:**
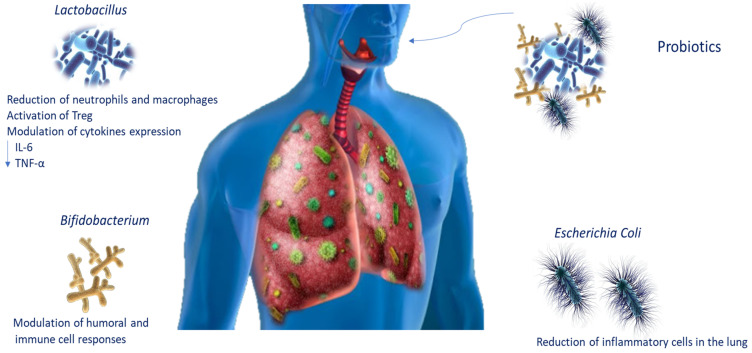
Probiotic modulation during lung infections. Schematic representation of probiotic modulation of immune responses during lung infections. Abbreviations: tumour necrosis factor-alpha (TNF-α); interleukin-6 (IL-6).

**Table 1 biomedicines-10-01337-t001:** Lung microbiota analysis in COPD patients.

Subject	Sample	Taxonomic Changes	References
218 COPD subjects	16S rRNA gene-based sputum microbiome	*Proteobacteria* *Haemophilus* *Moraxella*	[6]
281 COPD subjects	16S ribosomal RNA sputum samples	*↓Veillonella*	[41]
112 COPD subjects	Respiratory sample	*P. aeruginosa* was present in all COPD severity stages and colonisation	[45]
Healthy (*n* = 10), TSCOPD (*n* = 11), BMSCOPD (*n* = 10).	Nasal swabs and oral washings	*Actinomyces, Actinobacillus, Megasphaera, Selenomonas*, and *Corynebacterium* were significantly higher in COPD subjects	[47]
Healthy (*n* = 51) COPD (*n* = 70)	Clinical assessment and sputum induction.	↑*Haemophilus influenzae* detection was associated with higher sputum levels of NE and IL-1β, and *Streptococcus pneumoniae* was more common in male ex-smokers with emphysema and a deficit in diffusion capacity.	[48]
120 subjects COPD	Sputum	60% *H. influenzae*, 48% *M. catarrhalis*, 28% *S. pneumoniae*	[49]
78 COPD patients	Sputum investigated using 16S rRNA V3-V4 amplicon sequencing	↑*Gemella morbillorum*, ↑*Prevotella histicola*, ↑*Streptococcus gordonii*	[50]
200 severe COPD patients	In total, 1179 sputum samples were collected at stable, acute exacerbation, and follow-up visits.	*Moraxella, Haemophilus, Pseudomonas*, and *Staphylococcus* present in exacerbation events.	[51]

**Table 2 biomedicines-10-01337-t002:** The therapeutic effects of probiotics.

Probiotics	Therapeutic Effects	References
*Bifidobacterium*	Humoral and cellular immune responses modulation.	[91]
*Lactobacillus plantarum*	Decreases the number of macrophages and neutrophils and cytokine levels (IL-6 and TNF-α) to induce immunosuppression.	[92]
*Lactobacillus rhamnosus*	Regulates respiratory immune responses protecting from H1N1 influenza virus.	[93]
*Lactobacillus casei*	Stimulates immune cell migration inducing cytokine expression and decreasing pathogens. Increases natural killer cell activity in COPD smoker patients.	[94] [95]

## Data Availability

References for this review were identified through searches of PubMed for articles published from February 2002 to February 2022.

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
