# Peer review of "Impact of Lung Microbiota on COPD"

_biomedicines, 2022, doi:10.3390/biomedicines10061337_

Round 1
Reviewer 1 Report
The authors reviewed and summarized papers to describe the impacts of lung microbiota on COPD. I think this review paper is well-described, but a few points are to be addressed.
1. In the paragraph, “5. Lung microbial dysbiosis and COPD”, the authors mainly reveiwed papers focusing on lung microbiota patterns in patients with COPD or comparing that between COPD patients and healthy control subjects. Therefore, it could be hard to see if the findings were specific to COPD or not. So, It would be meaningful to review additionally if any specific dysbiosis pattern exists between COPD and other lung diseases such as interstitial lung diseases or pulmonary hypertension. That would help understand if the possible specific relationships between lung microbiota and COPD exist and could be a potential therapeutic target.
2. Regarding “Concluding remarks and future directions”, the description is redundant, therefore, should be shortened.
Author Response
Dear reviewer,
First of all, we would like to thank for reviewing the paper and suggesting improvements.
Please, find below a point-by point review of your valuable comments. We hope you are satisfied with our answers
- In the paragraph “Lung microbial dysbiosis and COPD”, we described dysbiosis in interstitial pneumonia as suggested, thank you. We have not reported the dysbiosis that affects other lung diseases (ie.: sarcoidosis, pulmonary hypertension) because as far as we know there is still not enough data to be reported
- Regarding the section “Concluding remarks and future directions”, has been shortened as you suggested. Thanks
Reviewer 2 Report
This review is certainly interesting because (except for the gut) little is known about native microbiota and its impact on (patho)physiological events in the human body. The manuscript is informative and from my point of view is suitable for the readership of the Biomedicines. However, I have several points that could improve the manuscript. An additional one or two figures/tables could be included to make the review easier to catch.
- The general information on the lung microbiota and its influence in lung diseases is based on some former reviews. A few recent reviews on the role of lung microbiota in COPD development should also be considered. Suggested references:
Liu, Jiexing, et al. "Role of pulmonary microorganisms in the development of chronic obstructive pulmonary disease." Critical reviews in microbiology 47.1 (2021): 1-12.
Invernizzi, Rachele, Clare M. Lloyd, and Philip L. Molyneaux. "Respiratory microbiome and epithelial interactions shape immunity in the lungs." Immunology 160.2 (2020): 171-182.
Caverly, Lindsay J., Yvonne J. Huang, and Marc A. Sze. "Past, present, and future research on the lung microbiome in inflammatory airway disease." Chest 156.2 (2019): 376-382.
- In Section 2 entitled Lung microbiota, the authors should have put a little more about fungal microbiota. In this respect, they could refer to a recent study by Yang et al. [Yang, Rui, et al. "Different Airway Inflammatory Phenotypes Correlate with Specific Fungal and Bacterial Microbiota in Asthma and Chronic Obstructive Pulmonary Disease." Journal of Immunology Research 2022 (2022).]
- When discussing exacerbations, the authors should also mention the association between airway inflammatory phenotypes and lung microbiota, which according to the level of peripheral blood and/or sputum eosinophil count, is divided into eosinophilic and noneosinophilic inflammatory phenotypes. In addition, a recent reference [Dicker, Alison J., et al. "The sputum microbiome, airway inflammation, and mortality in chronic obstructive pulmonary disease." Journal of Allergy and Clinical Immunology 147.1 (2021): 158-167.] identified microbiome-based subtypes of COPD associated with clinical phenotypes.
- Please more clearly present the taxonomic changes in the microbiota in Table 1.
- In Section 4, the additional Figure that shows ROS-induced cellular inflammatory response and oxidative damage could be very helpful for future readers.
- As the authors mentioned in the Concluding remarks and future directions of the cross-talk between the gut microbiota and the lungs, they should include an additional Section about the so-called gut-lung axis. Suggested references:
Bowerman, Kate L., et al. "Disease-associated gut microbiome and metabolome changes in patients with chronic obstructive pulmonary disease." Nature communications 11.1 (2020): 1-15.
Invernizzi, Rachele, Clare M. Lloyd, and Philip L. Molyneaux. "Respiratory microbiome and epithelial interactions shape immunity in the lungs." Immunology 160.2 (2020): 171-182.
- In Line 396, the authors should specify the probiotic strains of E. coli.
- In Section 7, the authors should consider adding a table that will summarize the therapeutic effects of probiotics.
I hope that the above-mentioned suggestions will strengthen the scientific value of the manuscript.
Author Response
Dear reviewer,
First of all, we would like to thank for reviewing the paper and suggesting improvements.
Please, find below a point-by point review of your valuable comments. We hope you are satisfied with our answers.
- Manuscripts on the role of lung microbiota in COPD development have been considered and the . suggested references: Liu, Jiexing, et al. " Critical reviews in microbiology 47.1 (2021); Invernizzi, R et al. Immunology 160.2 (2020) Caverly, Lindsay J., Yvonne J. Huang, and Marc A. Sze. "Past, present, and future research on the lung microbiome in inflammatory airway disease." Chest 156.2 (2019): 376-382, have been cited
- In Section 2 entitled Lung microbiota, some more information about fungal microbiota have been added and the recent study by Yang et al. Journal of Immunology Research 2022 has been cited as suggested
- The association between airway inflammatory phenotypes according to the level of peripheral blood and/or sputum eosinophil count,and lung microbiota has been deepened at page 9. The reference “Dicker, Alison J., et al. Journal of Allergy and Clinical Immunology 2021”, has been cited
- Table 1 has been corrected adding the taxonomic changes
- In Section 4, an additional figure showing ROS-induced cellular inflammatory response and oxidative damage has been created
- An additional Section about the so-called gut-lung axis has been written. The Suggested references: Bowerman, Kate L., et al. Nature communications 11.1 (2020); Invernizzi R et al Immunology 160.2 (2020): 171-182, have been cited as you suggested
- In line 396, probiotic strain of E. coli has been specified.
- A table which summing up the therapeutic effects of probiotics has been created as you suggested. Thanks
Reviewer 3 Report
The review article gives a comprehensive picture of lung microbiota and its association in context to COPD. I article is well structured and informative. Minor comments- How we can define good/bad microbiota, diverse/less diverse microbiota and associate this with COPD. May need some take home message on this point in Abstract/concluding remark.
Author Response
Dear reviewer,
First of all, we would like to thank for reviewing the paper and suggesting improvements. We hope you are satisfied with our answers.
In the section “Concluding remarks and future directions”, some sentence as take home message have been added as you suggested. Thanks
Round 2
Reviewer 1 Report
The authors responded well to my suggestions and the manuscript looks improved and more informative.
Reviewer 2 Report
The authors have satisfactorily addressed all my comments.